# Identifying Ovarian Cancer in Symptomatic Women: A Systematic Review of Clinical Tools

**DOI:** 10.3390/cancers12123686

**Published:** 2020-12-08

**Authors:** Garth Funston, Victoria Hardy, Gary Abel, Emma J. Crosbie, Jon Emery, Willie Hamilton, Fiona M. Walter

**Affiliations:** 1The Primary Care Unit, Department of Public Health and Primary Care, University of Cambridge, Cambridge CB1 8RN, UK; veh29@medschl.cam.ac.uk (V.H.); jon.emery@unimelb.edu.au (J.E.); fmw22@medschl.cam.ac.uk (F.M.W.); 2University of Exeter Medical School, University of Exeter, Exeter EX1 1TX, UK; G.A.Abel@exeter.ac.uk (G.A.); W.Hamilton@exeter.ac.uk (W.H.); 3Gynaecological Oncology Research Group, Division of Cancer Sciences, University of Manchester, Manchester M13 9WL, UK; emma.crosbie@manchester.ac.uk; 4Department of Obstetrics and Gynaecology, Manchester University NHS Foundation Trust, Manchester Academic Health Sciences Centre, Manchester M13 9WL, UK; 5Centre for Cancer Research and Department of General Practice, University of Melbourne, Melbourne, VIC 3000, Australia

**Keywords:** ovarian cancer, symptoms, early detection, risk assessment, diagnostic prediction model, triage tool, ovarian cancer symptoms

## Abstract

**Simple Summary:**

Most women with ovarian cancer are diagnosed after they develop symptoms—identifying symptomatic women earlier has the potential to improve outcomes. Tools, ranging from simple symptom checklists to diagnostic prediction models that incorporate tests and risk factors, have been developed to help identify women at increased risk of undiagnosed ovarian cancer. In this review, we systematically identified studies evaluating these tools and then compared the reported diagnostic performance of tools. All included studies had some quality concerns and most tools had only been evaluated in a single study. However, four tools were evaluated in multiple studies and showed moderate diagnostic performance, with relatively little difference in performance between tools. While encouraging, further large and well-conducted studies are needed to ensure these tools are acceptable to patients and clinicians, are cost-effective and facilitate the early diagnosis of ovarian cancer.

**Abstract:**

In the absence of effective ovarian cancer screening programs, most women are diagnosed following the onset of symptoms. Symptom-based tools, including symptom checklists and risk prediction models, have been developed to aid detection. The aim of this systematic review was to identify and compare the diagnostic performance of these tools. We searched MEDLINE, EMBASE and Cochrane CENTRAL, without language restriction, for relevant studies published between 1 January 2000 and 3 March 2020. We identified 1625 unique records and included 16 studies, evaluating 21 distinct tools in a range of settings. Fourteen tools included only symptoms; seven also included risk factors or blood tests. Four tools were externally validated—the Goff Symptom Index (sensitivity: 56.9–83.3%; specificity: 48.3–98.9%), a modified Goff Symptom Index (sensitivity: 71.6%; specificity: 88.5%), the Society of Gynaecologic Oncologists consensus criteria (sensitivity: 65.3–71.5%; specificity: 82.9–93.9%) and the QCancer Ovarian model (10% risk threshold—sensitivity: 64.1%; specificity: 90.1%). Study heterogeneity precluded meta-analysis. Given the moderate accuracy of several tools on external validation, they could be of use in helping to select women for ovarian cancer investigations. However, further research is needed to assess the impact of these tools on the timely detection of ovarian cancer and on patient survival.

## 1. Introduction

Ovarian cancer is the eighth most common cancer to affect women worldwide, accounting for over 384,000 deaths in 2018 [1]. Outcomes are strongly linked to stage at diagnosis, with five-year survivals of 90% and 4% for UK women diagnosed at stages I and IV, respectively [2]. Given this, large ovarian cancer screening trials have been conducted, but these have so far failed to demonstrate a significant reduction in long-term mortality [3,4]. In the absence of effective screening programs, the majority of ovarian cancers are diagnosed following symptomatic presentation [5,6], and a focus has been placed on the early detection of symptomatic disease [7].

While once regarded as a ‘silent killer’, many studies have demonstrated that a range of symptoms are more common in women with ovarian cancer than in control subjects and that symptoms occur at all stages of the disease [8]. Clinical guidelines in countries around the world recommend that patients presenting with symptoms of possible ovarian cancer undergo investigation, although debate remains over which symptoms are indicative of disease and should be included in guidelines [7]. To facilitate the early detection of symptomatic cancer, researchers have developed a number of symptom-based checklists for use either when patients first present in the clinical setting or in ‘symptom-triggered screening’ programs, in which symptoms are proactively solicited [9,10,11]. More sophisticated tools, which can take the form of diagnostic prediction models [12], have also been developed to incorporate test results and ovarian cancer risk factors alongside symptoms, in a bid to improve tool performance. Several of these tools have been incorporated into clinical computer systems, which, then, automatically alert the clinician to consider ovarian cancer investigations when relevant symptoms are present or when the risk of undiagnosed cancer reaches a certain level. However, the relative limitations and merits of the various available tools remain unclear. In this systematic review, we aimed to identify and compare the diagnostic performances of symptom-predicated tools for the detection of ovarian cancer.

## 2. Methods

### 2.1. Eligibility Criteria and Searches

This review was conducted and is reported in accordance with the Preferred Reporting Items for Systematic Reviews and Meta-Analysis (PRISMA) guidelines (Appendix A); a study protocol was registered with PROSPERO [CRD42020149879]. We searched MEDLINE, EMBASE and Cochrane CENTRAL for keywords relating to ovarian cancer, symptoms and prediction/diagnostic tools to identify papers published between 1 January 2000 and 3 March 2020 (Appendix A). The start date was chosen to predate the publication of key ovarian cancer symptom papers [13,14]. No language restrictions or restrictions on methodological design were applied. No restrictions were placed on study setting, so studies conducted in the general population or in primary, secondary, or tertiary care were all eligible for inclusion. Reference lists of included papers were screened to identify any additional relevant papers.

Studies were included if they (a) described the development and or evaluation of a multivariable tool designed to identify patients with undiagnosed ovarian cancer and (b) provided the sensitivity and specificity of the tool or gave sufficient data to allow these metrics to be calculated. For the purposes of this review, we defined a multivariable tool as a combination of three or more variables used to detect or predict the risk of undiagnosed ovarian cancer. This broad definition encompasses traditional multivariable diagnostic prediction models and clinical prediction rules [12,15]. We considered variable ‘checklists’, in which any one variable in the list needed to be present for a positive result, to be a form of multivariable tool. As the focus of this review was on symptom-based tools, the tool under investigation had to include at least one symptom for a study to be eligible. No other restrictions were placed on the type of variable that could be included in a tool. Studies on tools intended to estimate future risk of developing ovarian cancer rather than the current risk of having an undiagnosed ovarian cancer were excluded, as were studies on tools that solely provide an indication of the risk of relapse or recurrence. We excluded studies in which all participants had a pelvic mass—as this represents a highly selected high-risk population—and studies undertaken solely in paediatric (<18 years) populations. Non-primary research studies were also excluded.

### 2.2. Study Selection

The online Rayyan software was used to facilitate abstract screening and study selection [16]. Following removal of duplicates, two reviewers (G.F. and V.H.) independently screened titles and abstracts against eligibility criteria. Potentially eligible papers identified at the screening stage were obtained and the full texts were independently examined against eligibility criteria by two reviewers (G.F. and V.H.). Any disagreements were resolved by consensus.

### 2.3. Data Extraction and Synthesis

Data extraction was performed by one reviewer (G.F.) and checked against full-text papers by a second reviewer (V.H.) to ensure accuracy. Using a predeveloped template, information was extracted on study characteristics (year of publication and location), study design (methodology, population, data source and outcome definition), tools (variables and tool development methods), and tool performance metrics (sensitivity, specificity and other diagnostic metrics). Where a study evaluated multiple tools, data relating to each tool were extracted separately.

Sensitivity and specificity were used to compare tool accuracy. For diagnostic prediction models, area under the receiver operator characteristic curve (AUC) was used to compare discrimination (the ability of a tool to identify those with a condition from those without a condition) and calibration (agreement between estimated and observed outcomes). Due to the marked heterogeneity of included studies in terms of the study designs, populations, variable definitions, outcome definitions and use of different tool thresholds, and the failure of multiple studies to report numbers of patients with true positive/true negative/false positive/false negative results, we were unable to perform any meta-analyses. Instead, performance characteristics were summarised in tabular form and using a narrative synthesis approach. When synthesising data, we paid particular attention to several study and tool characteristics. First, the source of participant recruitment. For example, whether controls were recruited from the general population or after entry into healthcare, as symptoms may be more common in clinical controls than population controls, which could influence measures of tool sensitivity and specificity [17]. Second, whether the measures of tool accuracy were obtained directly from the patient sample in which the tool was developed (apparent performance), by applying internal validation methods, such as splitting the sample into development and validation sets or using cross-validation techniques (internal validation), or from a separate analysis in a distinct population (external validation) [12]. Tools usually exhibit poorer diagnostic performance in external validation studies than when evaluated in the original development sample, and external validation of tools is recommended before they are used in clinical practice [12]. Third, we considered whether tools consisted solely of symptoms or symptoms in addition to other variables, as this is likely to impact the clinical utility of the tool.

### 2.4. Risk of Bias Assessment

The Quality Assessment of Diagnostic Accuracy Studies 2 (QUADAS-2) tool was used to assess the risk of bias and applicability of the included studies [18]. QUADAS-2 includes signalling questions (intended to identify areas of potential bias or concern over study applicability) covering four domains: (1) patient selection, (2) index test(s), (3) reference standard and (4) flow and timing. Each domain was rated as having “high”, “low” or “unclear” (where insufficient information is provided) risk of bias. Domains 1–3 were also rated for applicability as “high”, “low” or “unclear” concern. Two reviewers (G.F. and V.H.) independently assessed each study using QUADAS-2. Ratings were compared and disagreements were resolved by consensus.

## 3. Results

### 3.1. Study Selection

In total, 2331 records were identified from database searches, of which 708 were duplicates. Two additional records were identified from examination of reference lists. A total of 1625 titles and abstracts were screened, and 35 full-text papers were examined. Sixteen studies met the eligibility criteria and were included (Figure 1).

### 3.2. Study Characteristics

The characteristics of the included studies are summarised in Table 1 and additional exclusion criteria are detailed in Appendix A. Three studies were population-based [19,20,21], five studies were based in a primary care setting [14,22,23,24,25], four studies were entirely hospital-based [26,27,28,29] and four studies were hospital-based but also recruited controls from screening studies [30,31,32,33]. All population- and hospital-based studies were of case-control design. Two of the studies that recruited from the hospital setting included a proportion of controls with benign ovarian pathology [26,28]. Three of the five primary care studies were of cohort design [22,23,24], and the remaining two were of case-control design [14,25]. The studies used a variety of data sources for variables, including pre-existing routinely collected primary care data (*n* = 6), information from surveys or patient interviews (*n* = 11) and blood samples (*n* = 4). Study sizes varied markedly, with 75–1,908,467 participants and 24–1885 women with ovarian cancer per study. While all studies used ovarian cancer as an outcome, how this was defined differed, with some only including invasive epithelial cancer or specifically stating that they excluded borderline tumours [19,20,21,26,27,28,29], and others apparently including both invasive and borderline epithelial tumours or all ovarian cancers [14,22,23,24,25,30,31,32,33]. One study included ovarian cancer alongside other common cancers in a composite outcome, but tool performance characteristics for each cancer were given separately [23]. Seven studies developed entirely new tools [14,19,22,23,25,30,33], six modified existing tools [26,27,28,29,31,32] and eight externally validated existing tools [20,21,24,26,27,28,29,33].

### 3.3. Risk of Bias

The main potential sources of bias were identified in the “patient selection” and the “index test” domains (Figure 2). As the case-control design can lead to overestimation of test performance [18], 13 studies were flagged as being at high risk of bias for patient selection. Key potential sources of bias identified for studies in the “index test” domain included failing to pre-define the tool threshold and retrospectively administering the tool after the outcome had been determined, e.g., questioning participants after the ovarian cancer diagnosis had been made. The risk of bias was generally judged as low for the “reference standard” and “flow and timing” domains. However, all primary care studies were flagged as being at high risk of bias in the “reference standard” domain as they relied on general practitioner (GP) records to identify ovarian cancer diagnoses, supplemented in two studies by death registration data [22,23] rather than hospital or cancer registry histological diagnoses. Concern over the applicability of studies was judged as low, save for the “reference standard” domain of one study which used a composite cancer outcome [23].

## 4. Tool Variables

The studies evaluated a total of 21 distinct tools, of which five were diagnostic prediction models developed using appropriate statistical methods from which variable weights were derived [12]. We grouped variables included in the tools into four categories: (1) patient demographics, (2) personal and family medical history, (3) symptoms and (4) test results (Table 2). By definition, all tools included symptoms, with 14 including only symptoms. Four tools incorporated demographics, two incorporated personal and family medical history and six incorporated test results. Five symptoms (abdominal pain, pelvic pain, distension, bloating and appetite loss) were included in more than half (≥11) of the tools and a further six symptoms (feeling full quickly, difficulty eating, postmenopausal bleeding, urinary frequency, palpable abdominal mass/lump and rectal bleeding) were included in at least a quarter (≥6) of the tools. Six tools were based on an existing tool—the Goff Symptom Index (SI)—which was modified to include additional symptom or test result variables. Specifications of each tool, including how variables were defined, are included in the Appendix A.

### 4.1. Evaluation of Tool Performance

The diagnostic performance of the included tools is summarised in Table 3. Measures of diagnostic performance for the majority of the tools were obtained directly from the patient sample with which the tool was developed (apparent performance) or by applying internal validation methods, such as splitting the sample into development and validation sets (internal validation), with only four tools—the Society of Gynaecologic Oncology (SGO) consensus criteria, Goff SI, QCancer Ovarian, Modified Goff SI 1—undergoing independent validation with an external dataset. Although the Goff SI in combination with CA125 was evaluated in several studies, the CA125 thresholds used varied markedly, so no studies were considered to have externally validated the same combination. There was overlap in evaluation of tools between healthcare settings, but no tool evaluated in primary care was evaluated in another setting or vice versa.

The most widely studied tool was the Goff SI, which was evaluated in nine studies [20,21,26,27,29,30,31,32,33], but two of these used data from subsets of women in the original tool development study [31,32]. Apparent deviations from the original Goff SI in how variables were defined were noted in several studies (Appendix A). The Goff SI was the only tool to be externally validated in groups of women recruited from more than one setting.

### 4.2. Tool Diagnostic Accuracy

#### 4.2.1. Hospital Setting

All but two tools evaluated in hospital populations incorporated the Goff SI. Two of these underwent external evaluation—the original Goff SI and a modified version incorporating additional symptoms (Modified Goff SI 1). The Goff SI, which was externally validated in six studies, demonstrated sensitivities which ranged from 56.9% to 83.3% (an outlier result) and specificities from 48.3% (an outlier result) to 98.9%. A modified version of the Goff SI (Modified Goff SI 1) demonstrated a sensitivity of 71.6% and a specificity of 88.5% in a single external validation study.

Augmenting symptom checklists with baseline risk factors and test results generally led to a reduction in sensitivity and an increase in specificity, or vice versa, depending on the threshold used. For example, the addition of the serum ovarian cancer biomarker CA125 to the Goff SI by Anderson et al. (2008) led to a reduction in tool sensitivity—if both variables were required to be abnormal for a positive tool result—or in tool specificity—if only one was required to be abnormal for a positive tool result [31].

#### 4.2.2. Population Setting

In women recruited from the population setting, two symptom checklists were externally validated side by side—the Goff SI and the SGO consensus criteria. While the sensitivities and specificities of the tools differed between the studies, within each study, they were similar, with an in-study maximum difference in sensitivity of 3.4% and specificity of 2.4% between the tools.

#### 4.2.3. Primary Care

A single tool (QCancer Ovarian), which took the form of a prediction model and combined symptom variables with demographics, family history and routine blood test results, underwent external validation in a primary care setting. When the threshold for abnormality was set to include the 5% of women at the highest predicted risk, QCancer Ovarian had a sensitivity of 43.8% and a specificity of 95%, while when the threshold was set to include women at the 10% highest risk, the sensitivity increased to 64.1% but the specificity fell to 90.1%. Several scores, developed by Grewal et al., demonstrated higher sensitivities and specificities than QCancer Ovarian at the 5% risk threshold (OC Score B ≥ 4) and 10% risk threshold (OC Score C ≥ 4), but diagnostic accuracy measures were derived from the same dataset used in score development.

Discrimination was reported for five tools (Table 3), all of which had similar AUCs within the ‘good’ range (0.84–0.89), with QCancer Ovarian exhibiting an AUC of 0.86 on external validation. Tool calibration was assessed for QCancer tools by graphically comparing the predicted cancer risk at two years with the observed risk by predicted risk deciles [22,23,24]. Authors reported good calibration on internal validation. On external validation, QCancer Ovarian had reasonable calibration but overpredicted risk, particularly in older women [24].

#### 4.2.4. Positive Predictive Values

The three cohort studies conducted in primary care reported positive predictive values (PPV) for QCancer tools at a range of thresholds (Table 3). The PPVs at any given risk threshold were similar—for example, values ranged from 0.5 to 0.8% when the threshold was set to identify the 10% of women at highest risk. Two case control studies (Rossing et al. and Jordan et al.) used external disease prevalence figures from screening studies and available population-level statistics to estimate the PPVs of the Goff SI and SGO consensus criteria—if they were to be used in general populations. The tools had similar estimated PPVs within each study, but PPVs were higher in Rossing et al. (0.63–1.12%) than in Jordan et al. (<55 years: 0.04–0.05%, ≥55 years: 0.18–0.31%).

## 5. Discussion

To our knowledge, this is the first systematic review to compare the diagnostic performance of existing symptom-based tools for ovarian cancer detection. We identified 21 symptom-based tools designed to help identify women with undiagnosed ovarian cancer. These tools comprised simple symptom checklists, checklists which included both symptoms and tests and more complex diagnostic prediction models which incorporated symptoms, test results and baseline risk factors. While the diagnostic performances of most tools were evaluated solely within the study development datasets, four tools were independently externally validated, with one being validated in multiple population settings. Externally validated tools demonstrated similar moderate diagnostic performances. Our findings should inform future studies evaluating the clinical impact of validated symptom-based tools when implemented in clinical practice.

### 5.1. Study Strengths and Limitations

The main strengths of this study were its systematic approach, broad search strategy and liberal eligibility criteria, which enabled us to identify and compare the performances of a wide variety of tools. However, the identified studies were extremely heterogeneous in their designs, populations, variable definitions, outcome definitions and thresholds, which ultimately precluded any meaningful meta-analyses. For example, although the Goff SI was evaluated in nine studies, there was overlap between the participants in three studies, control groups ranged from apparently healthy general population participants to hospital gynaecology patients (with or without benign pathology), ovarian cancer definitions differed and deviations in the parameters of the SI itself, in terms of symptom duration and frequency criteria, were noted in several studies. While meta-analysis was not deemed appropriate, our results demonstrate how the Goff SI performs under different conditions. An additional limitation was that all included studies were at high risk of bias in at least one QUADAS-2 domain, which limits the conclusions that can be drawn.

### 5.2. Comparison of Tools

Although all tools were symptom-based and designed to help identify women with ovarian cancer, they varied markedly in the symptoms they included. This mirrors discrepancies in the literature and within national guidelines as to which symptoms are associated with the disease and probably reflects differences in study methodologies and study populations [7]. Despite this, the symptoms with the highest positive likelihood ratios for ovarian cancer in a recent systematic review (distension, bloating, abdominal or pelvic pain) were incorporated into the majority of tools [8]. The more cancer-associated symptoms that are included in a checklist, the higher the sensitivity of the tool is likely to be, but at the cost of reducing specificity, as demonstrated by several of the included studies [19,26,33]. This was cited by Goff et al. as a rationale for not including urinary symptoms in the Goff SI [30]. Ultimately, variation in which additional symptoms a tool includes may have limited impact on tool performance; on external validation, two studies reported similar diagnostic accuracy metrics for the Goff SI and the SGO criteria (which differed on several symptoms), and on internal validation, Lim et al. concluded that changing several of the symptoms made relatively little difference to tool diagnostic accuracy [33].

In multiple studies, symptom checklists were augmented by ovarian cancer biomarkers with the aim of improving tool diagnostic accuracy. This approach naturally led to a reduction in tool specificity (where either symptoms or an abnormal test resulted in a positive tool) or sensitivity (where symptoms and an abnormal test were needed for a positive tool). If ovarian cancer biomarkers are to be included alongside symptoms within tools, this loss of performance could be avoided by incorporating them within prediction models, as per the inclusion of anaemia in QCancer Ovarian. As the prediction model threshold can be set at a desired risk level, biomarkers, such as CA125 and HE4, could be incorporated without harming tool performance. However, this would require women to have specialist ovarian cancer markers performed in order for the tool to be used, which significantly limits clinical utility. A more practical approach would be to incorporate tools within a two-step pathway in which symptom-based tools (which do not include specialist test variables) are used to help select higher-risk women for specialist ovarian cancer tests.

Variation in the reported sensitivity and specificity of the most widely evaluated tool, the Goff SI, was noted between studies. This variation is likely to be due, in part, to the marked differences in study design, populations and outcome definitions which precluded meta-analysis across these studies. Despite these differences, in 5 of the 6 external validation studies (including two large population-based studies), the Goff SI had a sensitivity in excess of 60%, and in all but the smallest study, which included only 24 ovarian cancers and 31 controls, its specificity exceeded 85%. The sensitivities and specificities of the two other externally validated symptom checklists—the SGO consensus criteria and the modified Goff SI 1—were similar, as were those of the only externally validated diagnostic prediction model—QCancer Ovarian (applying a 10% risk threshold). Given the similarity in performance of the various existing validated tools, future research efforts may be better directed at evaluating the impact of using available tools in practice rather than developing further tools consisting of different symptom combinations.

### 5.3. Clinical Relevance

Two distinct uses for tools were identified by the authors of the included studies: (1) assessment of women presenting symptomatically in the standard clinical setting to identify those at higher risk of undiagnosed cancer and to inform decision making and further investigation, and (2) proactive ‘symptom-triggered screening’ programs in which women are actively screened using the tool, with further testing for ovarian cancer occurring if the tool is positive. Several of the tools identified in this review are already available for use within the standard clinical setting in the form of electronic clinical decision support tools (eCDSTs). QCancer tools are integrated within some UK general practice IT systems and alert the clinician if the risk of ovarian cancer in an individual reaches a certain level, prompting them to consider ovarian cancer as a possible diagnosis. eCDSTs have been shown to improve practitioner performance and patient care, but there are multiple barriers to their implementation and they do not always lead to improved outcomes [43,44]. Therefore, even if eCDSTs are deemed to have acceptable diagnostic accuracy, their cost-effectiveness, acceptability to patients and clinicians and their impact on timely ovarian cancer detection and survival need to be evaluated. Currently, a large, clustered, randomised control trial is seeking to help to address this by investigating the clinical impact of implementing a suite of electronic cancer risk assessment tools (including an electronic version of the Hamilton ovarian SI) in UK general practice [45]. Studies have also sought to evaluate the impact of using tools as part of ‘symptom-triggered screening’ programs, but none have taken the form of randomised control trials—the gold standard approach—and so findings should be interpreted with caution. In one study, 5000 women were approached in primary care clinics and screened for symptoms using the Goff SI, with further investigations performed if the Goff SI was positive [11]. However, conclusions were limited as only two ovarian cancers were identified in the study window. The Diagnosing Ovarian and Endometrial Cancer Early (DOvEE) trial also employs a proactive symptom-triggered testing approach, supported by media campaigns, in which women can self-refer and are screened for range of symptoms prior to study inclusion. Although the final DOvEE results are yet to be published, a pilot study reported that participants had lower tumour burden and more resectable disease than women diagnosed via the standard clinical pathway [9].

When considering the clinical utility of a tool, it is important to assess the proportion of women who are ‘tool-positive’ who ultimately have ovarian cancer, i.e., the PPV. Primary care cohort studies indicated that between 1 in 200 and 1 in 100 women who were QCancer tool-positive (5% or 10% risk) had the disease. Although these figures may appear low, evidence indicates that patients would opt for cancer testing at PPVs of 1% [46]. Further, having a positive tool result in the clinical setting does not necessarily mean that further investigation will automatically occur, as there may be a clear alternative cause for the symptoms—the tool is simply intended as a diagnostic aid to highlight the risk of ovarian cancer to the clinician. In addition, the most common follow-up tests—CA125 and transvaginal ultrasound—are relatively non-invasive, and CA125 is known to perform well when used in a symptomatic primary care population [47], although invasive investigations/surgery may ultimately be needed to determine whether ovarian cancer is present. In proactive symptom-triggered screening programs, the tool is more than just a diagnostic aid—it is the initial screening step which will dictate whether further ovarian cancer tests take place. The two population studies reporting PPVs relied on external ovarian cancer prevalence figures, but their PPV estimates were similar to that reported in the pilot DOvEE study (0.76% in women ≥ 50 years) [9]. Further research is needed to help determine whether, given this PPV, follow-up testing in proactive symptom-triggered testing programs is acceptable to women and improves outcomes. The definitive diagnosis of ovarian cancer often involves invasive procedures/surgery, which has contributed to patient morbidity in key ovarian cancer screening trials [3,39]. Although initial findings indicate that proactive symptom triggered testing approaches lead to minimal unnecessary surgery [9,11], large trials are needed to confirm that the implementation of symptom-based tools in clinical practice does not lead to significant excess morbidity.

## 6. Conclusions

Over 20 symptom-based tools have been developed in different populations to help assess women for ovarian cancer, but the majority have not been validated. Four symptom-based tools—the Goff SI, a modified version of the Goff symptom Index, SGO consensus criteria and QCancer Ovarian—have undergone independent external validation and exhibit similar sensitivities and specificities. These tools could have an important role to play in the detection of ovarian cancer, but further large well-conducted studies are needed to assess their cost-effectiveness, their acceptability, their effect on the timeliness of ovarian cancer diagnosis and their impact on clinical outcomes, including patient survival.

## Figures and Tables

**Figure 1 cancers-12-03686-f001:**
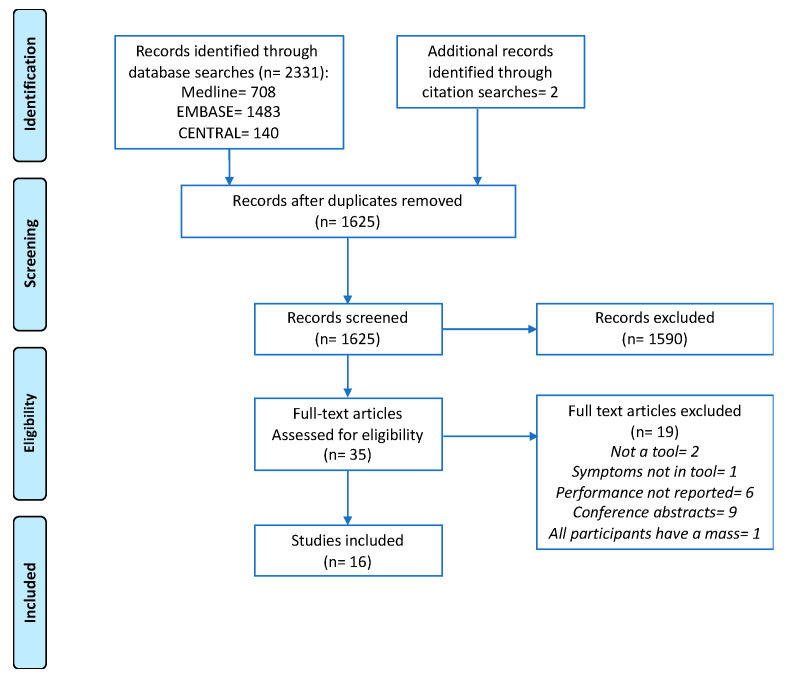
Preferred Reporting Items for Systematic Reviews and Meta-Analysis (PRISMA) flow diagram illustrating the study selection process.

**Figure 2 cancers-12-03686-f002:**
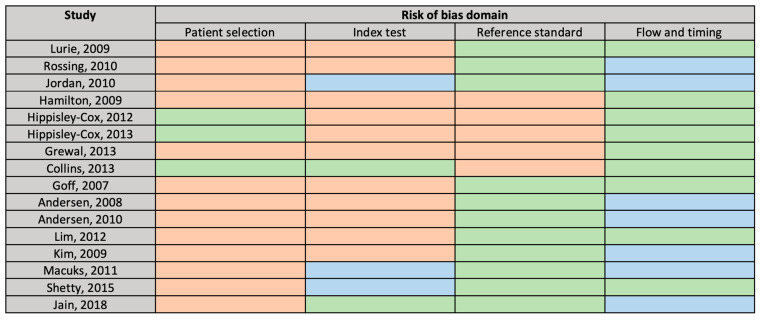
QUADAS-2 Risk of Bias Assessment. Green = “low”, orange = “high”, blue = “unclear” risk of bias.

**Table 1 cancers-12-03686-t001:** Study characteristics.

Author, Date, Country	Design		Objective			Primary Outcome	Candidate Variable Data Sources	Participants	Study Sample
	Case Control	Cohort	Develop New Tool	Modify Existing Tool	Externally Validate Existing Tool				
**Population based**									
Lurie, 2009, USA	●		●			Primary invasive ovarian carcinoma	In-person patient interviews using a structured survey	Cases: Women aged 19–88 years histologically-confirmed primary invasive ovarian carcinoma (1993–2007) Controls: Aged ≥ 18 years, Hawaii resident ≥ 1 year, randomly selected from statutory state survey Frequency-matched to cases (1:1) by age, ethnicity, interview time	Cases: 432 Controls: 491
Rossing, 2010, USA	●				●	Primary invasive epithelial OC ^a^	In-person interviews	Cases: Residents in western Washington State, aged 35–74 years, diagnosed with a primary invasive epithelial ovarian tumour (January 2002–December 2005) Controls: Selected by random digit dialling with stratified sampling in 5-year age categories, 1-year calendar intervals and two (urban vs. suburban or rural) county strata	Cases: 594 Controls: 1313
Jordan, 2010, Australia	●				●	Invasive epithelial OC	Patient survey	Cases: Aged 20–79 years with suspected OC, subsequently diagnosed with invasive epithelial OC (January 2002–June 2005) Controls: Frequency-matched based on age (5-year groups) and state of residence identified from electoral roll [34]	Cases: 1215 Controls: 1456
**Primary care population**									
Hamilton, 2009, England	●		●			Primary OC, including borderline	Researcher-coded GP records	Cases: Aged ≥ 40 years with primary OC diagnosed between 2000 and 2007 Controls: Matched on age, sex and GP practice	Cases: 212 Controls: 1060
Hippisley-Cox, 2012, England and Wales		●	●			OC (NOS)	QResearch database [35]	Aged 30–84 years, registered with GP practices between 1 January 2000 and 30 September 2010	Development (2/3)—1,158,723 women with 976 OCs Validation (1/3)—608,862 women with 538 OCs
Hippisley-Cox, 2013, England and Wales		●	●			OC (NOS) and 10 other cancers	QResearch database [35]	Aged 25–89 years, registered with GP practices between 1 January 2000 and 1 April 2012	Development (2/3)—1,240,864 women with 1279 OCs Validation (1/3)—667,603 women with 606 OCs
Grewal, 2013, England	●		●			Primary OC, including borderline	Researcher-coded GP records	Cases: Aged ≥ 40 years with primary OC diagnosed between 2000 and 2007 Controls: Matched on age, sex and GP practice	Cases: 212 Controls: 1060
Collins, 2013, UK		●			●	OC (NOS)	THIN database [36]	Women aged 30–84 years registered with GP practices between 1 January 2000 and 30 June 2008	1,054,818 women with 735 cancers
**Hospital + screening populations**									
Goff, 2007, USA	●		●			OC, including borderline	Patient survey	Cases: Women with a pelvic mass recruited in secondary care prior to OC diagnosis Controls: (a) Healthy ‘high-risk’ ^b^ women enrolled in a screening study [37], (b) women who presented for pelvic/abdominal US	Development Cases: 74 Controls: 243 Validation Cases: 75 Controls: 245
Andersen, 2008, USA	●			●		OC (NOS)	Patient survey, blood sample	Cases: Women with a pelvic mass, recruited prior to OC diagnosis Controls: Healthy ‘high risk’ ^b^ women enrolled in a screening study [37]	Cases: 75 Controls: 254
Andersen, 2010, USA	●			●		OC (NOS)	Patient survey, blood sample	Cases: Women with a pelvic mass recruited in secondary care prior to OC diagnosis Controls: Healthy ‘high risk’ ^b^ women enrolled in a screening study [37], frequency matched to cases on age (</>50 years)	Cases: 74 Controls: 137
Lim, 2012, UK	●		●		●	OC, including borderline	(a) Survey, (b) telephone interview, (c) GP notes	Cases: Women aged 50–79 years with primary OC recruited prior to diagnosis (February 2006–February 2008) Controls: Screening trial participants [38], frequency matched on year of birth and agreement to a telephone interview	Cases: 194 ^c^ Controls: 268 ^c^
**Hospital based population**									
Kim, 2009, Korea	●			●	●	Epithelial OC (NOS)	Patient survey, blood sample	Cases: OC diagnosis Controls: Women with benign ovarian cysts recruited prior to surgery and those undergoing routine pap smear	Cases: 116 Controls: 209 (Benign: 74, Pap smear: 135)
Macuks, 2011, Latvia	●			●	●	Epithelial OC (NOS)	Patient survey, blood sample	Cases: Women with epithelial OC recruited prior to surgery/diagnosis Controls: Age-matched ‘healthy women’ attending a gynaecology outpatient clinic ^d^	Cases: 24 Controls: 31 ^d^
Shetty, 2015, India	●			●	●	OC, excluding borderline	Patient survey	Cases: Women admitted to hospital for investigation and subsequently diagnosed with OC Controls: (a) Women with benign ovarian pathology; (b) those undergoing a ‘gynaecological check-up’	Cases: 74 Controls: 218 (benign: 144, gynaecological check-up: 74)
Jain, 2018, India	●			●	●	OC, excluding borderline	Patient survey, blood sample	Cases: Women undergoing surgery for a pelvic mass, subsequently diagnosed with ovarian cancer Controls: First-degree healthy relatives of cases	Cases: 45 Controls: 90

^a^ Data collected on borderline tumours but not included in their tool performance evaluation. ^b^ Women with high-risk family histories consistent with a possible BRCA1/2 mutation in their families, participating in the Ovarian Cancer Early Detection Study (OCEDS) [37]. ^c^ Numbers varied by study component: questionnaire (191 cases, 268 controls), telephone interview (111 cases, 125 controls) and GP notes (171 cases, 227 controls). ^d^ Controls with benign gynaecological disease were also included in study but are excluded from the review, as performance was examined separately to healthy controls and no overall specificity measure was given. Study design and Objectives denoted by “●”. Abbreviations: OC = Ovarian cancer; NOS = Not otherwise specified; GP = General practice; US = Ultrasound.

**Table 2 cancers-12-03686-t002:** Variables included in the final tools.

Tool (Study, Year)	Demographics		Personal/Family History		Symptoms													Test Results		
	Age	Other	PMH	FH	Abdo. Pain	Pelvic Pain	Increase Abdo. Size/Distens.	Bloat.	Appetite Loss	Feeling Full	Difficulty Eating	Weight Loss	Postmen. Bleeding	Rectal Bleeding	Palpable Abdo. Mass/lump	Urinary Freq.	Other	Hb	CA125	HE4
**Symptom checklists**																				
Goff SI (Goff, 2007)					●	●	●	●		●	●									
Modified Goff SI 1 (Kim, 2009)					●	●	●	●		●	●					●	Urinary urgency			
Lurie 7-SI (Lurie, 2009					●	●	●	●	●						●	●	Bowel symptoms, difficulty emptying bladder, dysuria, fatigue, abnormal vaginal bleed.			
Lurie 5-SI (Lurie, 2009)					●	●	●								●	●	Difficulty emptying bladder, dysuria, abnormal vaginal bleed.			
Lurie 4-SI (Lurie, 2009)					●	●	●								●		Abnormal vaginal bleed.			
Lurie 3-SI (Lurie, 2009)							●								●		Abnormal vaginal bleed.			
Hamilton SI (Hamilton, 2009)					●		●	●	●				●	●		●				
SGO consensus criteria * (Rossing, 2010)					●	●		●		●						●	Urinary urgency			
Lim SI 1 (Lim, 2012)					●	●	●	●	●	●		●			●					
Lim SI 2 (Lim, 2012)					●	●	●		●						●		Vaginal discharge			
Hippisley-Cox SI (Hippisley-Cox, 2012)					●		●		●			●	●	●						
Modified Goff SI 2 (Shetty, 2015)					●	●	●	●	●	●	●	●				●	Urinary urgency			
**Augmented symptom checklists**																				
Goff SI + CA125 (Andersen, 2008)					●	●	●	●		●	●								●	
Goff SI + HE4 (Andersen, 2010)					●	●	●	●		●	●									●
Goff SI + HE4 + CA125 (Andersen, 2010)					●	●	●	●		●	●								●	●
Goff SI + CA125 + menopause (Macuks, 2011)		Menopause			●	●	●	●		●	●								●	
**Prediction models**																				
QCancer Ovarian (Hippisley-Cox, 2012)	●			OC	●		●		●			●	●	●				●		
QCancer Female (Hippisley-Cox, 2013)	●	Townsend score, smoking, alcohol, BMI	T2DM, COPD, endomet. hyperplasia or polyp, chronic pancreatitis	OC, GI cancer, breast cancer	●		●		●			●	●	●			Difficulty swallowing, heartburn/indigestion, blood in urine, blood in vomit, blood when cough, irregular menstrual bleeding, vaginal bleeding after sex, breast lump, breast skin tethering or nipple discharge, breast pain, lump in neck, night sweats, venous thromboembolism, CIBH, constipation, cough, unexplained bruising	●		
OC Score A (Grewal, 2013)					●		●	●	●				●	●		●				
OC Score B (Grewal, 2013)					●		●	●	●				●	●		●				
OC Score C (Grewal, 2013)	●				●		●	●	●				●	●		●				

* Consensus statement released by the Society of Gynaecologic Oncologists (SGO), the Gynaecologic Cancer Foundation and the American Cancer Society. The presence of a variable within a model is denoted by “●”. The terms used to describe a given symptom varied subtly between studies—full details of each tool, including symptom terminology and duration and frequency criteria, are included in Appendix A. Abbreviations: PMH = past medical history; FH = family history; Abdo. = abdominal; Distens. = distension; Bloat. = bloating; Postmen. = postmenopausal; bleed. = bleeding; Freq. = frequency; Hb = haemoglobin; CA125 = cancer antigen 125; HE4 = human epididymis protein 4; SI = symptom index; OC = ovarian cancer; BMI = body mass index; endomet. = endometrial; T2DM = type 2 diabetes mellites; COPD = chronic obstructive pulmonary disease; GI = gastrointestinal; CIBH = change in bowel habit.

**Table 3 cancers-12-03686-t003:** Tool diagnostic accuracy.

Tool	Study	Recruitment				Source of Accuracy Estimate			Sensitivity (95% CI)	Specificity (95% CI)	PPV	AUC (95% CI)
		Population Level	1° Care	Hospital + Screening	Hospital	Apparent Performance	Internal Validation	External Validation				
**Symptom checklists**												
Goff SI	Goff, 2007			●			●		≥50 yrs: 66.7 <50 yrs: 86.7	≥50 yrs: 90 <50 yrs: 86.7	-	-
	Andersen, 2008 ^a^			●			●		64 (52.1–74.8)	88.2 (83.6–91.9)	-	-
	Kim, 2009				●			●	56.9	87.6	-	-
	Rossing, 2010	●						●	67.5 (65.4–69.6)	94.9 (93.9–95.8)	0.77–1.12 ^b^	-
	Jordan, 2010	●						●	68.1 (65.5–70.7)	85.3	0.09 ^c^ (≥55 yrs: 0.21–0.31 <55 yrs: 0.04) ^d^	-
	Andersen, 2010 ^a^			●			●		63.5 (51.5–74.4)	88.3 (81.7–93.2)	-	-
	Macuks, 2011				●			●	83.3	48.3	-	-
	Jain, 2018				●			●	77.8	87.8	-	-
	Lim, 2012			●				●	61.4–75.7 ^e^	89.6–98.9 ^e^	-	-
Modified Goff SI 1	Kim, 2009				●	●			65.5	84.7	-	-
	Shetty, 2015				●			●	71.6	88.5	-	-
7-symptom Index	Lurie, 2009	●				●			85	40	-	-
5-symptom Index	Lurie, 2009	●				●			80	63	-	-
4-symptom Index	Lurie, 2009	●				●			74	77	-	-
3-symptom Index	Lurie, 2009	●				●			54	93	-	-
Hamilton SI	Hamilton, 2009		●			●			85	85	-	-
SGO consensus criteria	Rossing, 2010	●						●	65.3 (63.1–67.4)	93.9 (92.8–95)	0.63–0.92 ^b^	-
	Jordan, 2010	●						●	71.5 (69–74.1)	82.9 (81–84.8)	0.08 ^c^ (≥55 yrs: 0.18–0.27 <55 yrs: 0.05) ^d^	-
Lim SI 1	Lim, 2012			●			●		69.6–91 ^e^	76–91 ^e^	-	-
Lim SI 2	Lim, 2012			●			●		67.3–91 ^e^	82.4–94 ^e^	-	-
Hippisley-Cox SI	Hippisley-Cox, 2012		●				●		71.9	82.9	0.5	-
Modified Goff SI 2	Shetty, 2015				●	●			77	88.5	-	-
**Augmented symptom checklists**												
Goff SI or CA125 ^f^	Andersen, 2008			●		●			89.3 (80.1–95.3)	83.5 (78.3–87.8)	-	-
Goff SI or CA125 (>35 U/mL)	Jain, 2018				●	●			97.8	68.9	-	-
Goff SI & CA125 (>21 U/mL)	Macuks, 2011				●	●			79.1	100	-	-
Goff SI & CA125 (>35 U/mL)	Macuks, 2011				●	●			70.8	100	-	-
Goff SI & CA125 (>65 U/mL)	Macuks, 2011				●	●			70.8	100	-	-
Goff SI or CA125 ^f^	Andersen, 2010			●		●			91.9 (83.2–97)	83.2 (75.9–89)	-	-
Goff SI or HE4 ^f^	Andersen, 2010			●		●			91.9 (83.2–97)	84.7 (77.5–90.3)	-	-
Any 1 of 3 (Goff SI or CA125 or HE4) ^f^	Andersen, 2010			●		●			94.6 (86.7–98.5)	79.6 (71.8–86)	-	-
Any 2 of 3 (Goff SI or CA125 or HE4) ^f^	Andersen, 2010			●		●			83.8 (73.4–91.3)	98.5 (94.8–99.8)	-	-
Goff SI & 1 or more of CA125 or HE4 ^f^	Andersen, 2010			●		●			58.1 (46.1–69.5)	98.5 (94.8–99.8)	-	-
Goff SI & CA125 (>25 U/mL) & menopause	Macuks, 2011				●	●			50	100	-	-
Goff SI & CA125 (>35 U/mL) & menopause	Macuks, 2011				●	●			45.8	100	-	-
Goff SI & CA125 (>65 U/mL) & menopause	Macuks, 2011				●	●			45.8	100	-	-
**Prediction models**												
QCancer Ovarian (Top 10% risk)	Hippisley-Cox, 2012		●				●		63.2	90.8	0.8	084 (0.83–0.86)
	Collins, 2013		●					●	64.1	90.1	0.5	0.86 (0.84–0.87)
QCancer Ovarian (Top 5% risk)	Hippisley-Cox, 2012		●				●		42.2	95.6	1.1	-
	Collins, 2013		●					●	43.8	95	0.6	-
QCancer Ovarian (Top 1% risk)	Hippisley-Cox, 2012		●				●		13.9	99.3	2.1	-
QCancer Ovarian (Top 0.5% risk)	Hippisley-Cox, 2012		●				●		11	99.6	3.2	-
QCancer Ovarian (Top 0.1% risk)	Hippisley-Cox, 2012		●				●		3.9	99.9	5.5	-
QCancer Female (Top 10% risk)	Hippisley-Cox, 2013		●				●		61.6	90	0.6	0.84 (0.82–0.86)
OC Score A (Score ≥ 3)	Grewal, 2013		●			●			58.5	97.3	-	0.89
OC Score A (Score ≥ 4)	Grewal, 2013		●			●			57.6	97.3	-	
OC Score B (Score ≥ 3)	Grewal, 2013		●			●			75	90.1	-	0.89
OC Score B (Score ≥ 4)	Grewal, 2013		●			●			58.9	97.3	-	
OC Score C (Score ≥ 3)	Grewal, 2013		●			●			85.4	85.1	-	0.88
OC Score C (Score ≥ 4)	Grewal, 2013		●			●			72.6	91.3	-	

^a^ Study used a subset of patients from Goff, 2007. ^b^ Calculated using external data from screening studies. [39,40]. ^c^ Calculated using external Australian population-level data. ^d^ Calculated using external data from US and UK screening studies and Australian population-level data. [41,42]. ^e^ Sensitivity and specificity varied by data collection method (questionnaire, telephone interview, GP notes). ^f^ Biomarker level (CA125, HE4) dichotomised at 95th percentile in control group—levels above that deemed abnormal. The Recruitment setting and the source of accuracy estimate are denoted by “●”. Abbreviations: OC = ovarian cancer; CI = confidence interval; AUC = area under the receiver operator characteristic curve; PPV = positive predictive values; yrs = years.

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
