# Peer review of "Identifying Ovarian Cancer in Symptomatic Women: A Systematic Review of Clinical Tools"

_cancers, 2020, doi:10.3390/cancers12123686_

Round 1
Reviewer 1 Report
This manuscript describes a systematic review of symptom-based tools to aid early detection of ovarian cancer. The methods are clearly-described, with a fairly comprehensive search strategy, and the involvement of two authors in selecting papers for inclusion and data extraction. The process of identifying relevant papers (and excluding non-relevant ones) is clearly shown in a flow diagram. The authors identified 16 studies evaluating 21 tools, and have summarised the characteristics of the different tools (e.g. the various component symptoms +/- risk factors or blood tests), and the results of the evaluations (e.g. study population, aim of evaluation, sensitivity, specificity) in a series of tables. The authors have discussed and summarised the results in narrative form, but have not performed a meta-analysis, as they judged (I think correctly) that there was too much heterogeneity between the studies for that to be appropriate.
I think this is a very strong paper, and I have very few suggestions for improvement. I think the methods are systematic, and the results are clearly described and thoughtfully discussed. Importantly, the authors highlight the key limitation of the existing literature: that there is very limited evidence that use of these symptom-based tools will lead to better outcomes in ovarian cancer (i.e. that using the tools(s) in a specific patient population leads to appropriate earlier diagnosis of ovarian cancer, with consequent reduced morbidity and mortality even after accounting for lead-time bias, and that this outweighs any harm from inappropriate excess investigation and treatment of women who do not have ovarian cancer). Even where specific intervention studies have been set up to evaluate the use of these symptom-based tools in clinical practice, they have tended not to be randomised controlled trials (e.g. the DOvEE study), which leads to difficulty interpreting the results, due to inevitable bias and confounding. This point could perhaps be made even more – but it is alluded to in the Simple Summary, Abstract, and Discussion, so the authors and editors may feel this is sufficient.
On a related note, although the authors suggest that “the most common follow-up tests – CA125 and transvaginal US – are relatively non-invasive” (Discussion, lines 403-404), the issue presumably is that definitive diagnosis of an ovarian mass often involve salpingoophorectomy, which is far from non-invasive, and carries substantial morbidity (and implications for fertility). While the risk of over-diagnosis in a symptomatic population will presumably be less than in an asymptomatic screening population, I think it is important to highlight this potential risk (which was certainly an issue in the trials of ovarian cancer screening, e.g. UKCTOCS and PLCO).
I have also noted a few possible typos, detailed below:
- Table 1: First page of table, last line (Jordan 2010) – I think ‘electoral role’ is perhaps meant to be ‘electoral roll’.
- Discussion, line 324: I think ‘An addition limitation was…’ should perhaps be ‘An addition[al] limitation was…’
- Discussion, line 409: I think ‘Further research is needed help determine whether…’ should be ‘Further research is needed [to] help determine whether…’
- Discussion, line 419: I think a comma is missing between ‘their acceptability’ and ‘their effect on the timeliness of ovarian cancer diagnosis’.
Overall, I thought this was an excellent paper. Although I have suggested some points to consider highlighting in the discussion, and a few typos, I have little in the way of substantial changes to propose.
Author Response
We are very grateful to the reviewer for taking the time to review the manuscript and provide helpful feedback. We have addressed each of the reviewers points -
Reviewer point 1: “Even where specific intervention studies have been set up to evaluate the use of these symptom-based tools in clinical practice, they have tended not to be randomised controlled trials (e.g. the DOvEE study), which leads to difficulty interpreting the results, due to inevitable bias and confounding. This point could perhaps be made even more…”
Author response: We have further highlighted that none of the published ‘symptom triggered testing’ studies have taken the form of randomised control trials, and that their findings should be interpreted with caution (page 21, lines 403-405).
Reviewer point 2: “While the risk of over-diagnosis in a symptomatic population will presumably be less than in an asymptomatic screening population, I think it is important to highlight this potential risk...”
Author response: We have added several lines to the Discussion section to highlight this potential risk (page 22, line 423-424 and 430-435).
Reviewer point 3: “I have also noted a few possible typos”
Author response: We have corrected these typos.
Reviewer 2 Report
This is the vast review of diagnostic performance of clinical tools based on symptoms to detect ovarian cancer. Studies were carefully selected to the analysis (16 of 2331). To compare the diagnostic accuracy sensitivity, specificity, PPV, ROC, AUC were used. The authors reviewed 21 symptom based tools. Only 4 of them vere validated and the authors concluded that might be usefull after large, well conducted studies.
Author Response
We are very grateful to the reviewer for taking the time to review the manuscript and for their feedback.
Reviewer 3 Report
1) Please expand on the use of symptom-predicted tools for the detection of ovarian cancer
2) Please expand on the discussion section to include additional information regarding the findings of the study and how this adds to the existing literature on symptom-based tools for ovarian cancer.
Author Response
We are very grateful to the reviewer for their feedback on the manuscript. We considered their comments, made minor revisions to the manuscript, and are confident that the topics highlighted by the reviewer have been covered in depth -
Reviewer point 1: “Please expand on the use of symptom-predicted tools for the detection of ovarian cancer”
Author response: In the Simple Summary (page 1, lines 21-23) Abstract (page 1, lines 33-34, 43-46) and Introduction (page 2, lines 63-66) we highlight that symptom-predicated tools have been developed to identify women at increased risk of ovarian cancer for further investigation. The ‘Clinical relevance’ section of the Discussion focuses on the clinical use of symptom-predicated tools. We discuss different clinical uses of the tools (page 21, lines 387-391), highlight where tools have already been implemented in clinical practice (page 21, lines 391-399) and discuss key studies evaluating tool use and clinical impact (page 21, lines 400-412). We have added a line to the Introduction, further highlighting how some tools are currently used in clinical practice (page 2, lines 69-71).
Reviewer point 2: “Please expand on the discussion section to include additional information regarding the findings of the study and how this adds to the existing literature on symptom-based tools for ovarian cancer”
Author response: We discuss the key study findings in several places in the Discussion section of the manuscript (page 20, lines 320-326, lines 345-363, page 21, lines 365-384). We highlight key implications of the study findings on research and clinical practice in the Discussion (page 21, lines 376-384) and in the Conclusion. We have added several lines to the Discussion section highlighting that this is the first review to systematically identify and compare the performance of symptom-based tools for ovarian cancer detection, and that the results will help inform future studies evaluating the impact of symptom-based tools implemented in clinical practice (page 20, lines 319-320 and 326-328).